# Resonant Scattering by Excited Gaseous Components as an Indicator of Ionization Processes in the Atmosphere

**Vasily Bychkov**

Institute of Cosmophysical Researches and Radio Wave Propagation, Far Eastern Branch of the Russian Academy of Sciences, Paratunka 684034, Russia; vasily.bychkov@ikir.ru

**Abstract:** The results of lidar sensing of the atmosphere at altitudes in the range of 25–600 km from the Kamchatka Lidar Station (53° N, 158° E) obtained in 2008–2022 are presented. The results of an analysis of the data of two-frequency lidar sensing of the thermosphere are given. The increased scattering at wavelengths of 532 and 561 nm is due to resonant scattering by excited atomic nitrogen and oxygen ions. Manifestations of resonant scattering in the middle atmosphere are also discussed. It is demonstrated that these ions are excited during the process of the ionization of the main atmospheric species by the precipitation of energetic electrons. The findings show that, during lidar soundings of the middle atmosphere, the ionization process can form imaginary aerosol formations. The spectrum of precipitating electrons can be estimated from the shape and position of the maximum of the lidar signal profile. It is shown that the process of the excitation of ions in the ground state does not play any significant role in the formation of the backscattered lidar signal. The signal does not carry information about the content and height profile of ions at the ground state. The appearance of resonant scattering in the atmosphere indicates the presence of the ionization sources.

**Keywords:** atmosphere; ionosphere; lidar; resonant scattering

## 1. Introduction

The method of the resonant scattering of laser rays propagating in the upper atmosphere was first proposed and implemented in 1964. The history of the development of the method is described in [1]. In this review article, the resonant scattering cross-sections were determined for the radiation transitions in He, NO, $N_2^+$, and $N_2$ atmospheric gases to be in the wavelength range from 300 to 1100 nm and for the transitions in Na, K, Li, and Ca metal atoms and ions in the range of visible light. For the metal ions, the resonant scattering cross-section has an order of $10^{-13}$ cm$^2$, which is 13–14 orders of magnitude higher than that of the molecular scattering cross-sections of the main atmospheric gaseous components. The resonant scattering cross-sections of the gaseous components for different transitions lie in the range from $10^{-12}$ cm$^2$sr$^{-1}$ to $10^{-21}$ cm$^2$sr$^{-1}$. High values of scattering cross-sections make it possible to receive lidar signals returned from the thermosphere. The first experiments on the laser sensing of atmospheric sodium at altitudes in the range of 80–100 km were carried out in 1968 [2]. The first experimental measurements of the horizontal wind velocity and the temperature based on the resonant scattering data at altitudes in the range of 80–100 km were carried out in the same years.

Most currently published work has been based on metal ion scattering due to the large cross-sections and their frequent occurrence at altitudes from 80 to 140 km. Lidar systems have already been developed for the measurement of the ion content, the temperature, the wind velocity and their dynamics, under variable geophysical conditions [3–6]. In [7], a method for remotely measuring the magnetic field at the altitude of the thermosphere was proposed.

In terms of lidar studies of the main gaseous components of the thermosphere, there are several works which should be mentioned, including a lidar designed for the study

of the excited states of ionized molecular nitrogen [8,9]. In that study, they presented the expected backscattered signals calculated for a telescope with a mirror that was 1 m in diameter. Based on the study of possible transitions between the excited states of the ionized molecular nitrogen, the wavelengths of 390.303 nm were proposed for the laser. A wavelength of 391.537 nm was chosen to record scattering signals. A light filter with a bandwidth of 0.3 nm was used. The expected backscattered signal profiles were calculated up to an altitude of 300 km. The Kamchatka lidar station, equip ped with a Brilliant-B (Quantel, France) laser (532.08 nm) and a Newton telescope (IOA SB RAS, Tomsk, Russua)with mirror that was 60 cm in diameter, came into operation in the autumn of 2007 for the investigation of aerosols at altitudes from 25 to 80 km. The background noise signal was calculated in the upper-range gates of signals backscattered at altitudes above 100 km. On some days, the lidar signals revealed an *anomalous* behavior: the total lidar signal remained tilted to the vertical axis up to 500–600 km. The background noise signal and possible effects of afterpulses of a photoelectron multiplier were investigated in March and September 2008. A correlation was found between the signal integrated in the range of heights of 200–300 km and the critical frequency foF2 of the F2 layer of the ionosphere [10,11].

Figure 1 illustrates an example of the correlation during observations on 28March 2008. Figure 1a shows variations of 15 min values of the critical frequency foF2 obtained from ionosonde measurements. The critical frequency foF2 is proportional to the electron density $(Ne)^{1/2}$ in the maximum of the F2 layer. Figure 1b shows a plot of the backscattered signal S accumulated over 15 min and integrated over the 200–300 km layer. Similar correlations were observed during the autumn equinox on 5–6 September 2008 [11].

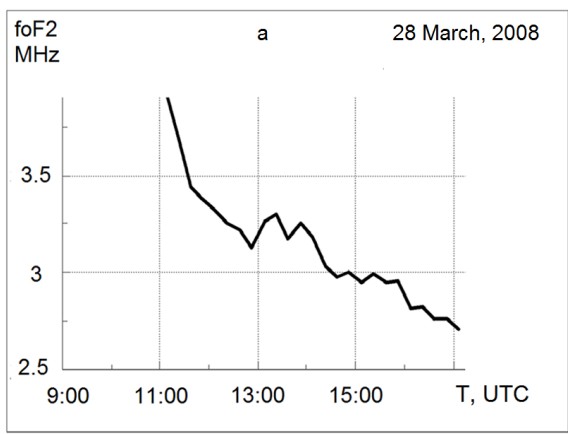 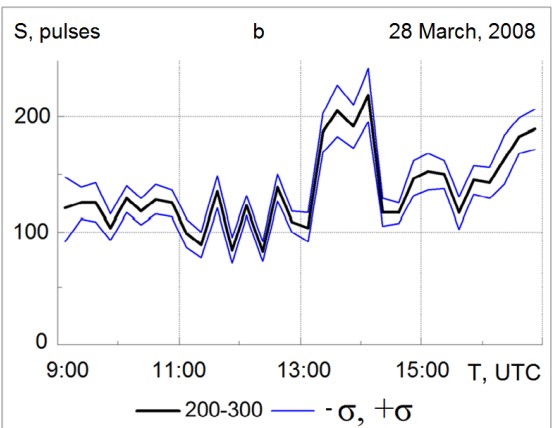

**Figure 1.** Temporal variations (**a**) of the critical frequency foF2 of the ionospheric F2 layer and (**b**) of the backscattered lidar signal observed at altitudes from 200 to 300 km [10].

The correlations of the scattering ratio R averaged over ~5 km layers, with the ionospheric parameter $f_{min}$, were also found from lidar data obtained in January–February 2008 [12] (see Figure 2). The scattering ratio is determined as R = (βa + βm)/βm, where βa and βm are the aerosol and molecular backscattering coefficients. The R(H) values greater than 1 indicate the occurrence of aerosol formations at the altitude H. This issue will be discussed in more detail in Section 4.3. The parameter $f_{min}$ is the minimal frequency at which the E or F layer trace appears in the ionograms. If $f_{min}$ is higher than the frequency $f_o$ at which the session of ionospheric sensing starts, the radio waves are absorbed in the frequency range $(f_o - f_{min})$.

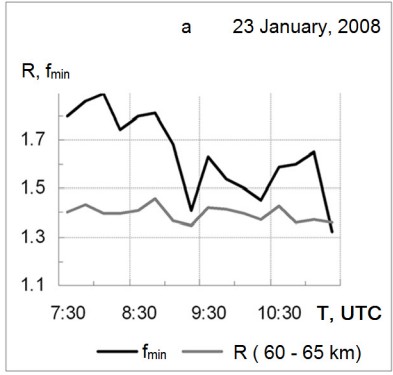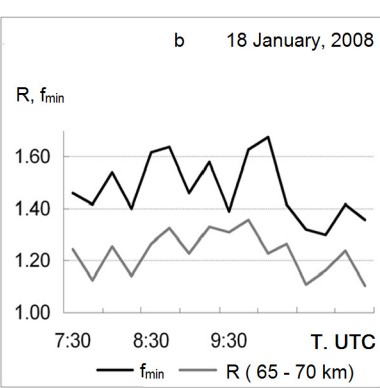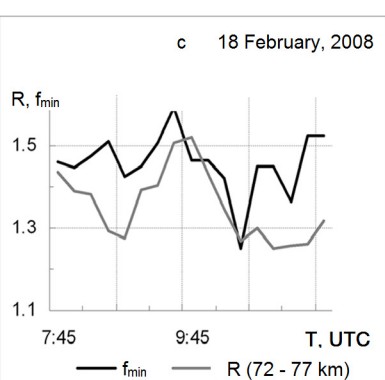

**Figure 2.** Variations of the layer-averaged scattering ratio R (gray curve) and ionospheric parameter $f_{min}$ (black curve) during different observational intervals: (**a**) on 23 January 2008; (**b**) on 18 January 2008; (**c**) on 19 February 2008 [12].

The electron oscillates with the electromagnetic wave frequency, and the moving electron radiates at the same frequency. The absorbed energy returns to the wave. The energy is lost when electrons collide with neutrals. The expression for the wave attenuation coefficient has the form $\chi \sim \nu(\omega_0^2/\omega^2)$. The attenuation is proportional to the frequency of electron collisions with the neutrals and the squared plasma frequency (or the plasma density) and inversely proportional to the squared frequency of the incident radio wave. In the Earth's atmosphere, the attenuation of the sounding wave occurs mainly at altitudes in the range of 75–90 km. At lower altitudes, the electron density decreases, and at higher altitudes, the density of the neutrals decreases. The increase inthese ionospheric parameters leads to the corresponding increase in the free electron content in the F2 layer maximum (foF2) and in the mesopause ($f_{min}$).

The results obtained suggest that resonant scattering is observed. The lines of excited nitrogen ions were detected in the laser radiation spectrum. After the modernization of the lidar with a dye laser, the line indicative of excited oxygen atom ions at 561.107 nm was chosen as the most efficient in terms of the laser pulse energy. In the present paper, we use the spectral lines of excited oxygen and nitrogen ions in order to analyze the ionization of the upper atmosphere.

## 2. Instrument and Method

During observations in 2017, we used the two-frequency lidar with an Nd:YAG laser (Brilliant-b, Quantel, France) to generate radiation at a wavelength of 532 nm and a dye laser (TDL-90, Quantel, France) to generate radiation at a wavelength of 561 nm, with a pulse repetition frequency of 10 Hz. The main parameters of the lidar used in our experiments are given in Table 1. The optical block diagram of the lidar is shown in Figure 3.

**Table 1.** Equipment.

| Transmitter 1 | Transmitter 2 | Receiver |
|:---:|:---:|:---:|
| Brilliant-B Nd:YAG Laser Pulse energy—400 mJ Wavelength—532.08 nm Line width—0.040 nm Pulse duration—5–6 ns Beam divergence—0.5 mrad | TDL-90 dye laser YG-982E pump laser Pulse energy—100 mJ Wavelength—561.106 nm Line width—0.025 nm Pulse duration—10 ns Beam divergence—0.5 mrad | Telescope mirror diameter—60 cm H8259-01 Hamamatsu PMT M8784-01 photon counters Vertical resolution—1.5 km Light filter bandwidth—1 nm |

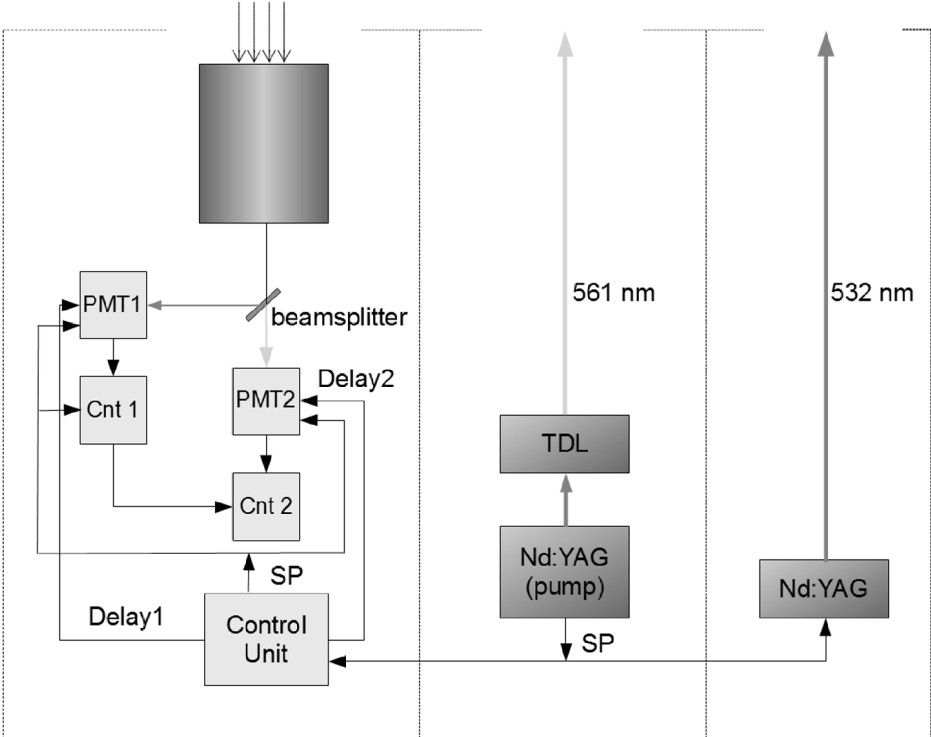

**Figure 3.** Optical block diagram of the lidar system. Here, TDL is the dye laser, NdYAG is the solid-state laser (532 nm), SP is the synchronizing pulse, Cnt 1 and 2 are photon counters, PMT indicates the photomultiplier tube, and CU is the control unit.

The pump laser (YG-982E, Quantel, France) forms a synchronizing impulse which is transmitted to the second solid-state laser, the Tunable Dye Laser (TDL), and the control unit (CU). In turn, the CU transmits the synchronizing pulses to two photomultiplier tubes (PMTs) and the photon counting boards. Blocking pulses of a preset duration are simultaneously formed and transmitted to both PMTs to protect them from illumination by signals from the near zone.

To avoid PMT overloading due to contamination by signals from the near zone, electronic PMT locking with a 140 μs pulse was used in both receiving channels. This corresponds to an elimination of the data from the nearest ~25 km. The received signals were stored as binary files with 10 s of data accumulation, which allowed their subsequent summation with arbitrary time periods to be carried out. Typically, they were set equal to 15 min in accordance with the ionozonde operation mode. Measurements of the background noise signal started 20 ms after the transmission of each laser pulse and lasted for 4 ms with a 10 μs step. Thus, the background noise did not contain afterpulses, and the data accumulation rate was good.

Information about the ionospheric conditions is obtained from measurements of the "Parus A" ionozonde. The sounding is conducted in the standard mode of once per 15 min. The ionozonde and lidar operation were controlled by computers synchronized in time by GPS. Each session of ionospheric sensing lasted 20 s. The program (developer IKIR FEB RAS, 2008) for lidar data preprocessed deleted signals obtained during ionozonde operation to exclude the possible effects of spurious signals by the receiving lidar system.

In 2017, sensing was performed at wavelengths of 532.08 and 561.107 nm. The dipole transitions at these wavelengths are given in Table 2 [13]. Here, OII is the $O^+$ ion, NII is the $N^+$ ion, NIII is the $N^{++}$ ion, and Aik are the Einstein coefficients defining the transition probability. Figure 4 shows the location of the excited ion lines in the spectra of the lasers from Table 2. Section 4.1 provides an energy level diagram for all states of $O^+$ and $N^+$ ions, transitions which are allowed from the upper-level state of Table 2.

**Table 2.** Dipole transitions of excited oxygen and nitrogen ions falling within the emission bands of the lasers.

| Component | Wavelength (nm) | Aki (s⁻¹) | Lower Level | Term | J | Upper Level | Term | J |
|---|---|---|---|---|---|---|---|---|
| O II | 561.1061 | $2.14 \times 10^6$ | $2s^22p^2(^1S)3s$ | $^2S$ | 1/2 | $2s^22p^2(^3P)4p$ | $^2P°$ | 1/2 |
| NIII | 532.0870 | $5.68 \times 10^7$ | $2s2p(^3P°)3p$ | $^2D$ | 5/2 | $2s2p(^3P°)3d$ | $^2F°$ | 7/2 |
| NII | 532.0958 | $2.52 \times 10^7$ | $2s2p^2(^4P)3p$ | $^5P°$ | 1 | $2s2p^2(^4P)3d$ | $^5P$ | 2 |

The location of resonance lines (Table 2) in the emission spectra of lasers is shown in Figure 4.

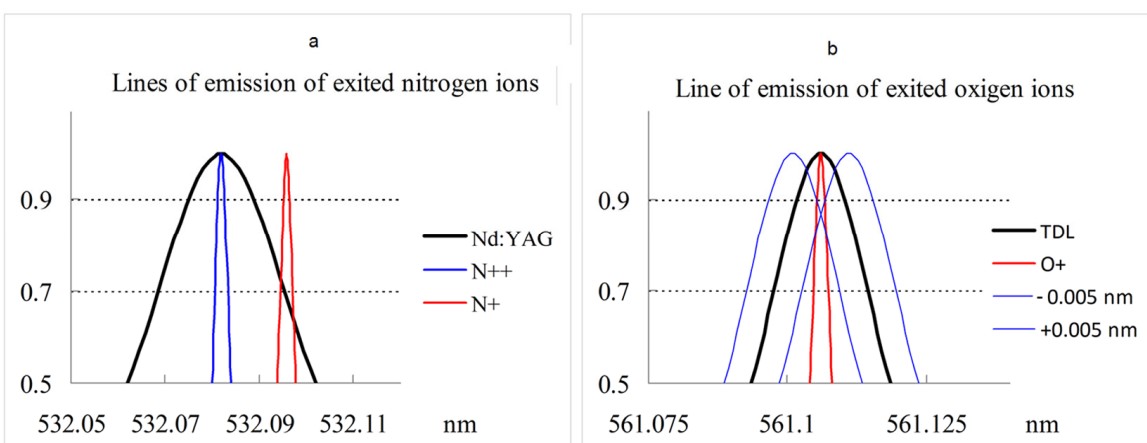

**Figure 4.** Emission lines of excited nitrogen (**a**) and oxygen (**b**) ions in the emission spectra of the Nd:YAG and TDL lasers.

The lines shown in Figure 4 and listed in Table 2 were chosen with an allowance for the laser emission bandwidth and a Doppler line broadening equal to ~0.004 nm at ionospheric altitudes and a temperature of 800 K. The doubly ionized nitrogen ion line falls at the center of the laser emission band, but the content of these ions is low. The main contribution to the lidar return signal is provided by the N⁺ ions.

The dashed curves on the right side of Figure 4 show the shift of the laser emission spectrum relative to the oxygen ion emission line due to a possible error in the calibration of the TDL laser. It is assumed that the laser wavelength is set to an accuracy of 0.01 nm.

## 3. Experimental Data

In the introduction, we presented measurements of resonant scattering in the thermosphere and mesosphere (Figures 1 and 2) at a wavelength of 532 nm. The total signal strengths that were received from the thermosphere and accumulated for 15 min were ~(1–3) × 10². Several cases of backscattering in both lidar channels were observed between August and November 2017. A characteristic feature of the data obtained in this period was the presence of scattering at 200–400 km, which was absent at altitudes in the range of 100–200 km. Moreover, the total lidar signal strength at altitudes ranging from 200 to 400 km exceeded, by an order of magnitude, the corresponding values observed in 2008.

The data obtained on 5 and 23 September 2017 (Figures 5 and 6) are used in this work. Two events were observed on 23 September. All other events in the thermosphere were visually similar. However, resonant scattering from the thermosphere, mesosphere, and stratosphere was observed on 5September. In all cases, the geomagnetic conditions were quiet during lidar observations. According to the data obtained from the geomagnetic Paratunka Observatory, local three-hour K-indices were equal to (2 3 2 2 2 1 2 1) on 5 September and (1 1 1 0 2 1 2 1) on 23September. On 23 September, the background noise signals accumulating for the oxygen and nitrogen lines for 15 min were 21–25 and 9–10 ADC units, respectively. On 5 September, the background noise signal recorded for

the oxygen line was 80–100 ADC units, which increased toward the end of the observations; for the nitrogen line, it was 25–27 ADC units.

Figures 5 and 6 show the spatiotemporal distributions of the lidar return signals observed on 5 and 23 September 2017 after the subtraction of the background noise signal. The sliding average method was used, with a window of 4.5 km, to smooth the profile with altitude. The signal values were multiplied by the k × H$^2$ coefficient, where H is the altitude and k = $10^{-4}$ [14]. The total lidar signal S that accumulated overnight and the same signal Sn, normalized by the k × H$^2$ coefficient, are shown at the bottom of the figure.

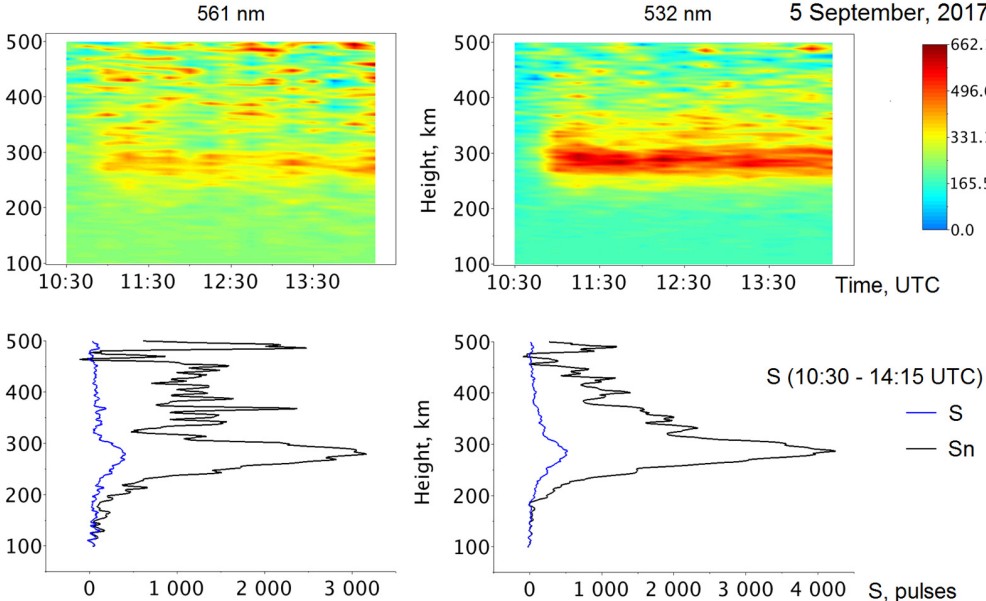

**Figure 5.** Spatiotemporal distribution of the lidar signals with background noise subtracted for an altitude in the range of 100–500 km (at the **top**) and the total signal S (blue curve) and normalized signal Sn (black curve) (at the **bottom**), obtained on 5 September 2017.

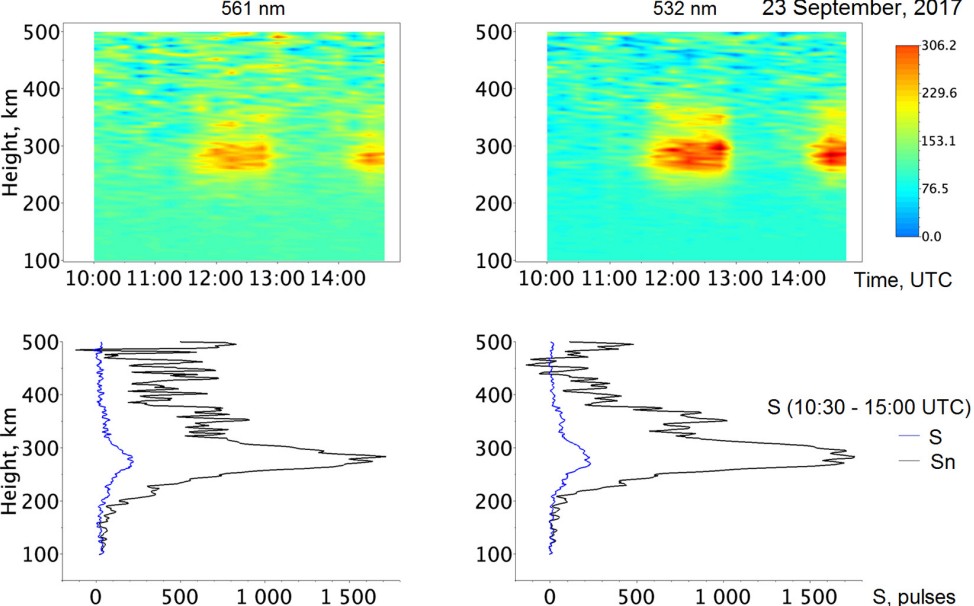

**Figure 6.** The same as that in Figure 5, but for 23 September 2017.

Normalization by the squared altitude corresponds to the actual decrease in the true signal with increasing altitude. It was assumed that aerosol and molecular scattering were absent at altitudes above 100 km. With a chosen coefficient value of k = $10^{-4}$, the signal at an altitude of 100 km coincided with the actually detected signal, thereby improving visual data perception.

Figure 7 shows the plots of the lidar return signal averaged over a layer of 200–400 km and the plots of the foF2 and foEs critical frequencies observed on 5 and 23 September 2017 [15]. Here, Figure 7a,c at the top show the lidar data after a 15 min accumulation, with the background noise signal subtracted. The total signals obtained on 23 September at wavelengths of 532 and 561 nm are well correlated and almost identical. On 5 September, the signal values at wavelengths of 561 and 532 nm were well correlated, but they were 20–40% less at the wavelength of 561 nm than those at the wavelength of 532 nm. The critical frequencies of the F2 and Es layers during lidar observations are shown at the bottom of Figure 7. The time of the occurrence of enhanced scattering from the ionospheric F2 layer is indicated in red. One can clearly see that the increase in the foF2 critical frequency occurs simultaneously with the increase in the lidar signal. The 0.1–0.3 MHz increase in the critical frequency of the F2 layer at night and the presence of corpuscular type layers among the Es layer frequencies at night observed on 5 September confirm the assumption of the occurrence of superthermal (0.1–10 keV) electron precipitation in the ionosphere.

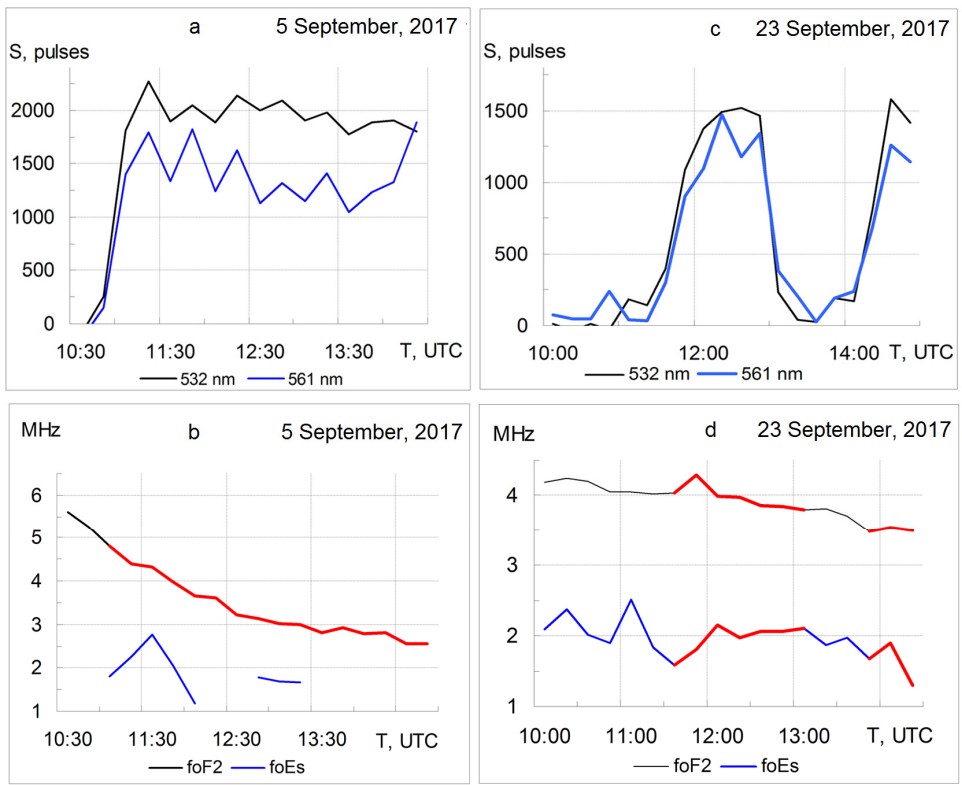

**Figure 7.** Lidar signal with background substracted summed over a 200–400 km layer (**a**,**c**) and the critical foF2 and foEs frequencies (**b**,**d**) during lidar observations on 5 and 23 September 2017.

## 4. Discussion and Main Results

The main features noted in the lidar sensing data obtained from August–November 2017 are as follows:

1.  It was expected that the value of a lidar signal at a wavelength of 561 nm would be several times higher than at a wavelength of 532 nm, since the content of $O^+$ ions at altitudes in the range of 150–400 km is about two orders of magnitude higher than that of the $N^+$ ions [16]. Our lidar observations showed that the total overnight signal

values at a wavelength of 532 nm were typically 20–30% higher than those obtained at a wavelength of 561 nm.

2.　The altitude of the maxima of the scattered signal did not coincide with that of the F2 layer maximum. The lidar signal peaked at 280–290 km in altitude. According to the ionosonde data obtained on 5 September 2017, the F2 layer maximum was located at altitudes of 300–350 km when the scattering layer maximum was observed.

3.　Enhanced light scattering was observed with the second local maximum at altitudes of 350–450 km (Figures 5 and 6) for all lidar signals at the wavelength of 561 nm. In all cases, the signal at the wavelength of 532 nm decreased monotonically from its maximal value with increasing altitude.

These features of the vertical lidar signal profiles have little in common with the structures of the vertical profiles of the oxygen and nitrogen atomic ions in the ground state. It thus follows that the process of the excitation of the existing ions plays no significant role in the formation of resonant scattering.

The main properties of the ionosphere used are detailed below. The most important for the formation of the ionosphere are the reactions:

$$
\begin{array}{ll}
N_2 + h\nu > N^+_2 + e, & N_2 + e > N^+_2 + 2e, \\
O_2 + h\nu > O^+_2 + e, & O_2 + e > O^+_2 + 2e, \\
O + h\nu > O^+ + e, & O + e > O^+ + 2e.
\end{array}
\tag{1}
$$

Only the first step upon ionization by UV radiation from the Sun involves the participation of photons. All subsequent ionization steps occur mainly with the participation of secondary electrons. The results of the modeling of the ionosphere at altitudes in the range of 100–400 km are presented in [16]. The model parameters were corrected using measurements from the Atmosphere Explorer C satellite. A schematic representation of the photochemical processes and the rates of component formation in the lower thermosphere were given in that work. The most important results for us are summarized below:

The ionization of molecular nitrogen is the determining process for the occurrence of $N^+$ ions in the ground state.

Among the main processes for the formation of $O^+(^4S)$ ions in the ground state are the ionization of molecular and atomic oxygen.

The rate of the creation of the $O^+$ ions is higher, by a factor of approximately 1.5, than that of the $N^+$ ions.

More than half of all $N^+$ and $O^+$ ions created are in excited states.

*4.1. Mechanism for Formation of the Resonant Scattering Signal*

The resonant scattering signal is formed through the absorption of the laser photon by the ion in the lower-level state (see Table 2) and then from the subsequent emission due to a transition from the upper level at the same frequency. The lower-level states, as given in Table 2, are also excited. The excited ion lifetime $\tau$ is determined by:

$$
\tau = 1/(\Sigma A_{ik} + \nu),
\tag{2}
$$

where $\nu$ is the frequency of ion collisions with neutrals, and the Einstein coefficients are summed over all states in which radiative transitions are possible [17].

Above 100 km, we can set $\nu = 0$, since, at these altitudes, the frequency of ion collisions is $\nu < 5 \times 10^3 \ s^{-1}$, which is much less than the radiative transition frequencies. A search for all such states in the NIST database [13] gives $\tau$ values equal to 1.06, 1.42, and 12.82 ns for $O^+$, $N^{++}$, and $N^+$ ions, respectively. These values have the same orders of magnitude as the pulse duration $T_{pulse}$ at wavelengths of 532 and 561 nm, equal to 5 and 10 ns, respectively.The total vertical profiles of the lidar signals (Figures 5 and 6) corresponding to the vertical profiles of the ionization rate by a monoenergetic electron beam presented in [18]. The calculations in [18] are confirmed by experimental data.

The laser pulse interacts with the excited ions in a thin layer during the time $T_{pulse}$. The ions created in this layer during the time $T_{pulse}$ participate in the interaction. The initial content of the excited ions in any thin layer of the atmosphere should be proportional to the rate of the creation of these ions multiplied by the lifetime τ of this state. The signal should be proportional to the ion creation rate multiplied by the sum of the pulse duration and the excited ion lifetime.

In the first approximation, the signal can be estimated as:

$$N \sim V \times (\tau + T_{pulse}) \times p, \tag{3}$$

where p is the probability of ion interaction with the laser pulse photons.

The portion of the laser pulse energy contributing to the resonant scattering can be estimated by taking the ratio of the excited transition line half-width to that of the laser radiation spectrum (Figure 4). With the allowance for Doppler line broadening equal to ~0.004 nm for both ions and the data presented in Table 1, this portion will be ~10% at a wavelength of 532 nm and ~15% at a wavelength of 561 nm. Hence, the effective pulse energy is about 40 and 15 mJ at wavelengths of 532 and 561 nm, respectively.

Let us estimate the probability of the interaction of an ion with a photon created in the layer during the time when the laser pulse acts on the layer. In the classical approximation, the resonant scattering cross-section can be represented in the following form:

$$\sigma = 3/2\pi \times \lambda^2, \tag{4}$$

where λ is the wavelength [19]. By substituting wavelengths of 532 and 561 nm, we obtain $1.35 \times 10^{-13}$ m$^2$ and $1.5 \times 10^{-13}$ m$^2$, respectively.

The photon energies are equal to 2.33 and 2.3 eV or $3.72 \times 10^{-19}$ and $3.68 \times 10^{-19}$ J at wavelengths of 532 and 561 nm, respectively. Taking into account the data presented in Table 2 and shown in Figure 4, we obtain the numbers of effective photons Np per pulse, which are equal to ~$1.0 \times 10^{17}$ and $0.4 \times 10^{17}$ at wavelengths of 532 and 561 nm, respectively. The collimators employed reduced the beam divergence (Table 1) by 7–8 times, down to ~$7 \times 10^{-5}$ rad. At an altitude of 300 km, the illuminated surface area S will be equal to 350 m$^2$.

The pulse energy loss due to molecular scattering in the atmosphere (20%) and on the output mirror surface and four collimator lens surfaces (10% on each surface) is 55%. At analtitude of 300 km, 0.45·Np/S photons pass by each square meter of the sounding surface, whose total numbers are equal to $1.28 \times 10^{14}$ and $5.1 \times 10^{13}$ at wavelengths of 532 nm of 561 nm, respectively.

The probability of interaction with the created ions is determined by multiplication with the number of photons Nτ that have passed through the unit area during the ion lifetime τ and the scattering cross-section σ rather than by the total number of effective photons per pulse. If Nτ×σ is greater than unity, then the probability of interaction is set to be equal to 1; otherwise, it is set to be equal to Nτ×σ.

At wavelengths of 532 and 561 nm, Nτ is respectively equal to $1.28 \times 10^{14}$ or $5.1 \times 10^{12}$ because the nitrogen ion lifetime is greater than the pulse duration, and for the oxygen ions, it is equal to 0.1·$T_{pulse}$. Comparing these values to the scattering cross-sections of $1.35 \times 10^{-13}$ and $1.5 \times 10^{-13}$ m$^2$, respectively, we find that the probability that the N$^+$ ions interact is close to unity. For the O$^+$ ions, the interaction probability is ~0.75. At an altitude of 200 km, the sounding surface area is halved, and the probability of the O$^+$ ion interaction for the entire underlying thermosphere becomes close to unity. For convenience, these data are given in Table 3.

**Table 3.** Main parameters of radiation interaction with the excited ions.

| Component | $\tau_{life}$, ns | $\sigma$, m$^2$ | Np | N$\tau$ | N$\tau \times \sigma$ | Interaction Probability P |
|---|---|---|---|---|---|---|
| N$^+$ | 12.82 | $1.35 \times 10^{-13}$ | $1.0 \times 10^{17}$ | $1.28 \times 10^{14}$ | 17.2 | 1 |
| O$^+$ | 1.06 | $1.5 \times 10^{-13}$ | $0.4 \times 10^{17}$ | $5.1 \times 10^{12}$ | 0.75 | 0.75 |

When calculating the interaction probability, the decrease in the probability of the interaction of the ions created before the pulse leaves the examined layer when the interaction time becomes less than the ion lifetime should be taken into account. For the O$^+$ ions, it decreases from 0.75 to 0 in the last 1.06 ns, and for N$^+$ ions, it decreases from 1 to 0 during T/(N$\tau \times \sigma$) = 0.28 ns. A similar situation is observed for the ions contained in the pulse that initially enters the layer, except that the probabilities increase from zero to those indicated in column 7 of Table 3.

Considering that many of the parameters used for the estimates are approximate, such corrections are small enough that they can be considered insignificant. The estimate obtained suggests that, at a wavelength of 532 nm, the pulse energy is redundant in terms of the interaction with all excited N$^+$ states created at the altitude of the F2 layer maximum.

The values of (T$_{pulse}$+$\tau$) at wavelengths of 532 and 561 nm are equal to 17.8 and 11 ns, respectively, with the allowance for corrections and interaction probabilities of 17.5 and 7.5 ns. Considering that the O$^+$ and N$^+$ ion creation rates are equal, the N$_{561}$/N$_{532}$ ratio of the signals is equal to 7.5/17.5 = 0.42. Hence, each pair of laser pulses from the lidar excites 2.5 times more nitrogen ions to the upper level than it does oxygen. The experimentally obtained signal ratio was N$_{561}$/N$_{532}$~0.8.

In the resonant absorption of laser pulse photons, oxygen and nitrogen ions participate in excited states at the lower level, as shown in Table 2. The rates of their formation depend on the ionization rate and can be estimated from experimental data. Expression (3) determines the number of acts of the resonant absorption of laser pulse photons with a transition to the upper level. The lidar signal is determined by the fraction of ions that return to their original lower-level state with emissions at wavelengths of 561.107 and 532.0958 nm. To estimate this fraction, it is necessary to find all states for which transitions from the upper level are possible and then estimate the probability of a transition using the values of the Einstein coefficients. These probabilities can be calculated for each ion by solving a system of differential equations for the rates of change in the population at all levels radiatively linked with the upper level, as in Table 2. However, this calculation is difficult.

In the 50–1500 nm wavelength range, we find 8 nitrogen lines and 10 oxygen lines, corresponding to the transition from the upper level, as shown in Table 2 [13]. The energy level diagram of these transitions is shown in Figure 8. It can be assumed that the proportion of top-level ions in Table 2 for each allowed transition will be proportional to the contribution of the Einstein coefficient to the total sum that determines the lifetime of this state. Then about 2.5% of the oxygen ions and ~0.6% of the nitrogen ions participate in the transition for radiation at resonant frequencies. Taking into account the probabilities of transitions with resonant emissions, there should be a reduction in the number of nitrogen photons by a factor of 2.5/0.6 ~ 4, giving the ratio N$_{532}$/N$_{561}$ = 2.5/4 = 0.625. From the lidar data for the rates of appearance of the lower-level states given in Table 2, we obtain V$_N$/V$_O$ = 1/0.625/0.8 = 2.

In the altitude range from 200–400 km, the atmospheric species are distributed at heights in accordance with its own altitude scale (mg/kT). The content of atomic oxygen varies from 50% at 200 km to 90% at 400 km. The ionization cross-section is larger for the molecular components. The molecular nitrogen and oxygen contentsare ~50% and ~2%, respectively, at an altitude of 200 km and ~7% and ~0.1% at an altitude of 400 km [20]. The appearance rate ratio obtained for the lower-level states shown in Table 2 is realistic and can explain the observed signal ratio at these two wavelengths. The further refinement of this relationship does not affect the results and is beyond the scope of the paper.

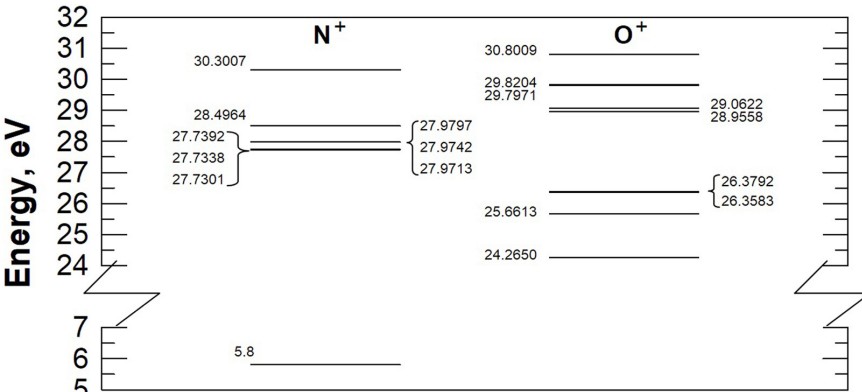

**Figure 8.** The energy level diagram for the states of O$^+$ and N$^+$ ions, transitions which are possible from the upper-level state of Table 2.

### 4.2. Estimated Spectra of Precipitated Particles

Figure 9a shows the vertical profile of the lidar signal, and Figure 9b shows the vertical profiles of the rate of ionization by precipitated electrons calculated from the data presented in [21]. The ionization rate calculated from analytical approximations [21] is in good agreement with the data presented in [18].

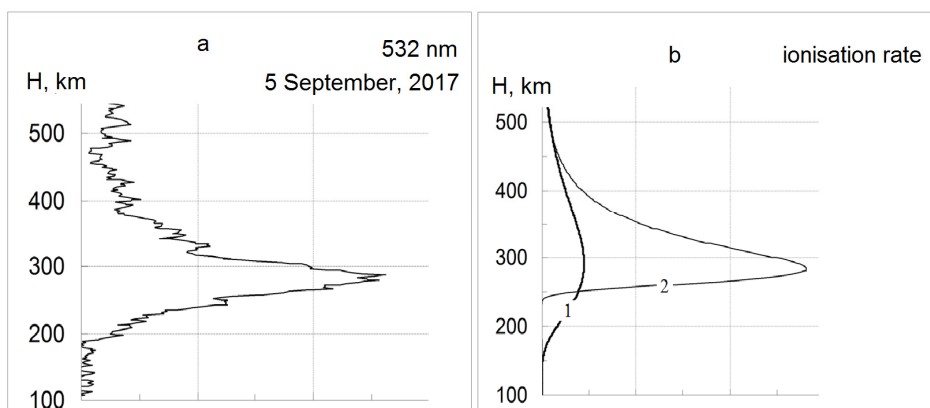

**Figure 9.** Vertical profiles of the lidar signal (**a**) and ionization rate for the Maxwell (curve 1) and mono-energy (curve 2) spectra of precipitated electrons (**b**).

The vertical profile of the ionization rate q(H) shown in Figure 9 was calculated for the Maxwellian spectrum of electrons with a characteristic energy of 120 eV (curve 1) and for a mono-energy beam of 330 eV electrons (curve 2). The electron energy was chosen such that the altitude of the location of the maximum ionization rate corresponded to that of the maximum signal (280–290 km). The concentrations of neutral $N_2$, O, and $O_2$ species were determined using the NRLMSISE-00 model [20]. The electron flux was $J_0 = 10^8$ cm$^{-2}$s$^{-1}$.

The vertical lidar signal profile in Figure 9a was derived from the nitrogen ion scattering data obtained on 5 September 2017, coinciding with the profile shown in Figure 5. The second maximum in the vertical oxygen ion scattering profile arose due to the ionization of atomic oxygen. The excited atomic oxygen ion states have a lifetime of 1 ns, and we do not have to worry about transfer from other regions. The atomic oxygen content at an altitude of 400 km was about $1 \times 10^8$ cm$^{-3}$ and accounted for 90% of the total content of the main atmospheric species. The ionized oxygen O$^+$ content was lesser by four orders of magnitude ($<10^4$ cm$^{-3}$). According to [22,23], the O ionization cross-section and the O$^+$ excitation cross-section should be the same or close in terms of the order of magnitude. In the creation of excited O$^+$ ions, the process of O ionization at this altitude should be much more important than the process of the excitation of the existing ions. The excitation of O$^+$ and N$^+$ ions does not play a significant role in the formation of lidar signals at any altitude.

### 4.3. Resonant Scattering in the Middle Atmosphere

The scattering ratio $R = (\beta_a + \beta_m)/\beta_m$ is widely used in lidar studies of the middle atmosphere, where $\beta_a$ and $\beta_m$ are the aerosol and molecular backscattering coefficients, respectively [24]. An R(H) value greater than 1 indicates the occurrence of aerosol formations at altitude H. The results of lidar observations in 2007–2012 established the following seasonal peculiarities in the occurrence of aerosol formations in the middle atmosphere of Kamchatka [25]:

In the stratosphere, aerosol formations arise during the cold season from November to March, and they are absent during the warm season from May to September.

In the mesosphere, aerosol formations appear during all seasons.

The conditions in the middle atmosphere are not favorable for water vapor crystallization or aerosol formation. The temperature and the water content in the mesosphere were estimated in [12] based on data from the Aura meteorological satellite and lidar observations in the presence of aerosol scattering. It was shown that, even at extremely low temperatures, as observed from 2007 to 2009, the water content in the mesosphere was more than an order of magnitude lower than that required for the crystallization of water vapor and the formation of aerosols. In the same work, strange correlations of the lidar signal (532 nm) averaged over 5–6 km layers in the mesosphere at altitudes of 60–77 km with the ionospheric parameter $f_{min}$ were observed in January–February 2008 (Figure 2). When analyzing the lidar data, we used the results of measurements of electron fluxes on the Demeter satellite, which flew at an altitude of 660 km both to the east and west of Kamchatka during lidar observations. The relativistic fluxes of electrons with an energy above 100 keV were recorded by the satellite. From the spectrum of electrons recorded at a latitude of 53° N on 18 January 2008, the ionization rate was calculated using analytical approximation, as proposed in [21]. They showed that such fluxes can cause ionization, reaching a maximum at an altitude of about 75 km.

Figure 10 shows vertical profiles of the scattering ratio R at altitudes in the 30–80 km range obtained on 5 September 2017. We can note the following main features:

The vertical profiles shown in Figure 10a,b were obtained without precipitation in the thermosphere (see Figure 5) and, as a whole, correspond to the vertical profiles observed in September.

The vertical profiles shown in Figure 10c,d did not appear in warm season observations in 2007–2017.

The vertical profiles of the scattering ratios for aerosol scattering at close wavelengths should be similar in form. In contrast, no similarity is observed between the profiles shown in Figure 10c,d.

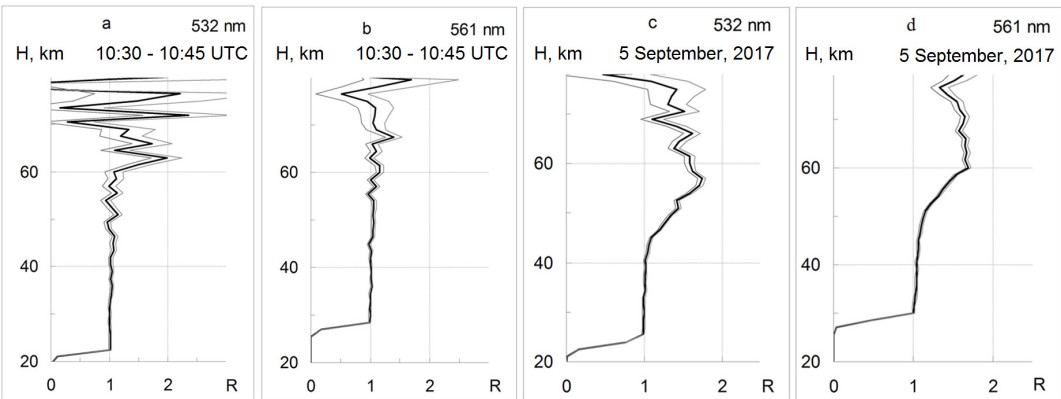

**Figure 10.** Vertical profiles of the scattering ratio of lidar signals recorded on 5 September 2017: (**a**,**b**) before the occurrence of a light-scattering layer in the thermosphere at 10:30–10:45 UTC; and (**c**,**d**) overnight accumulation of the scattering ratio on 5 September 2017. The thin curves indicate the standard deviations of the scattering ratio.

The conditions for the occurrence of resonant scattering in the middle atmosphere remained. The broadening of the Doppler line (~0.002 nm) at a temperature of 200° K (in the mesopause) is still sufficient for the formation of resonant scattering conditions. Atomic nitrogen and oxygen are almost absent in the middle atmosphere. With the presence of ionization sources, more than 50% of ions are created in excited states. The main sources of these ions are molecular oxygen, nitrogen, and ozone. The lifetime of excited ions in the mesosphere is still much less than the frequency of ion collision with neutrals, and the photon flux density at these altitudes is higher by an order of magnitude than that at an altitude of 300 km. The excited nitrogen and oxygen ions created in this altitude range interact with the photons with a probability which is close to 1. Hence, the detection of aerosol formation on 5 September 2017 (Figure 10) should be recognized as spurious. The maxima in the vertical R profiles at altitudes ranging from 55 to 75 km are actually formed by precipitating electrons with energies of 150–600 keV.

In the mesosphere, the frequency of ion collisions with neutrals can be calculated with the formula:

$$\nu(N) = 0.81 \times 10^{-10} \, (T/M)^{1/2} N \, (s^{-1}), \tag{5}$$

where T is the temperature, M is the molecular mass, and N is the concentration of atmospheric particles [17]. Figure 11a shows the vertical profiles of the excited nitrogen and oxygen ion lifetimes, and Figure 11b,c shows the signals of resonant scattering by nitrogen and oxygen ions. The probability of the interaction of ions created in a given range gate with laser pulse photons was calculated in [26]. They found that, at altitudes from 40 to 10 km, the probability of interaction was close to unity and increased with decreasing altitudes.

At an altitude of 10 km, the lifetime of excited ions is determined by ion collisions with neutrals, and it is close to ~0.2 ns. The photon flux density at an altitude of 10 km is higher by three orders of magnitude than that at an altitude of 300 km. Hence, the flux energy is more than sufficient, and, thus, the ion lifetime had practically no effect on the value of the signal. The signal is proportional to the pulse duration and the ionization rate. The ionization rate of oxygen increases significantly with the occurrence of ozone, as shown in Figure 11. It should be noted that the resonant scattering signal of 22,000 counts at an altitude of 35 km (see Figure 11c) is equal to about 1% of the total signal at the same altitude and cannot be distinguished in the vertical profile of the scattering ratio.

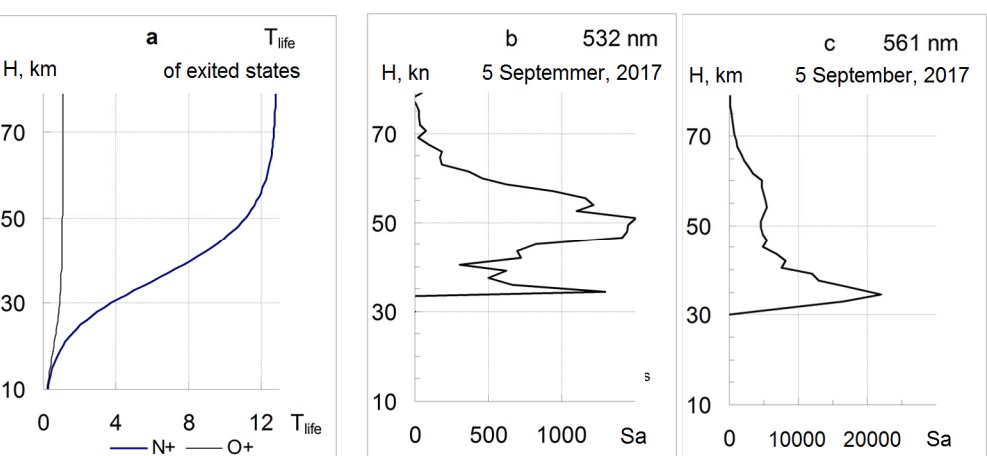

**Figure 11.** Vertical profiles of the lifetimes of the lower-level states N$^+$ and O$^+$ ions (see Table 2) (**a**); obtained on 5 September 2017 in the middle atmosphere signal Sa determined as: Sa = "the total backscattering signal" minus "the molecular scattering signal" minus "the background noise signal", 532 nm (N$^+$) (**b**); 651 nm (O$^+$) (**c**).

### 4.4. Observations in 2021–2022

During the years from 2021 to 2022, lidar sensing was performed only at a wavelength of 532 nm. In January–February of the both years, the occurrence rate of light-scattering layers was quite similar to those shown in Figures 5 and 6, but the layers were lower in intensity by one order of magnitude. An example of these observations is shown in Figure 12.

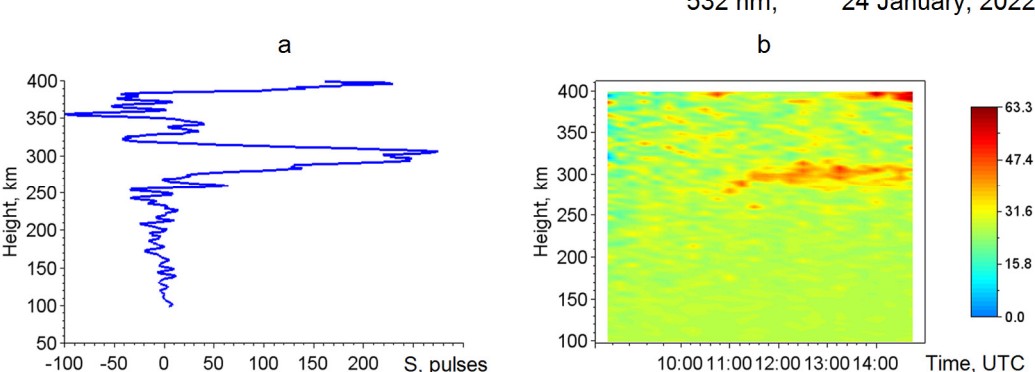

**Figure 12.** The total normalized signal Sn (**a**), spatiotemporal distribution of the normalized lidar signals with background noise subtracted for an altitude in the range of 100–500 km (**b**) obtained on 24 January 2022.

In all cases, the scattering appeared during the recovery phase of geomagnetic storms, as clearly explained in [27]. Specifically, there was an intensive precipitation of energetic electrons during the storm ring current decay and a restoration of the geomagnetic field. The coincidence of the energies of electrons precipitated in 2021–2022 and 2017 confirms the assumption that this process is controlled by the Earth's magnetic field. The higher-intensity events of 2017, which occurred underquiet geomagnetic conditions during the warm season, require further analysis.

### 5. Conclusions

We performed an analysis of the lidar data and geophysical conditions accompanying the occurrence of light-scattering layers in the atmosphere from lidar sensing data. Conclusions were made based on resonant scattering, and a mechanism for the interaction of laser pulse photons with excited ions was proposed. The following conclusions can be reached:

Resonant scattering at wavelengths of 532.08 and 561.107 nm occurs because of excited atomic nitrogen and oxygen ions created by the ionization of oxygen, nitrogen, ozone, and atomic oxygen molecules.

The process of atomic oxygen ionization can be manifested through resonant scattering at altitudes above 300 km. At lower altitudes, it is masked by the contribution of the molecular component.

The effect of the excitation of $O^+$ and $N^+$ ions in the ground state is insignificant. The lidar data do not contain significant information about their concentration in the atmosphere.

Resonant scattering in the middle atmosphere can result in the occurrence of spurious aerosol formations. The conditions for the manifestation of resonant scattering are retained throughout the entire altitude range for the application of lasers with pulse energies of 100–200 mJ.

A method for estimating the spectra of electrons precipitating in the atmosphere was proposed.

**Funding:** The research was carried out as a part of implementation of the Russian Federation state assignment AAAA-A21-121011290003-0.

**Institutional Review Board Statement:** Not applicable.

**Informed Consent Statement:** Not applicable.

**Data Availability Statement:** Common Use Center "North-Eastern Heliogeophysical Center CKR_558279, UNU 351757" http://www.ikir.ru/en/CUC/ (1 January 2023).

**Conflicts of Interest:** The author declares no conflict of interest.

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
