# Peer review of "Resonant Scattering by Excited Gaseous Components as an Indicator of Ionization Processes in the Atmosphere"

_atmosphere, doi:10.3390/atmos14020271_

Round 1
Reviewer 1 Report
This paper gives very interesting results, showing the lidar scattering signal of 200 to 400 km, and gives a theoretical explanation. This is very significant for improving lidar detection altitude and studying the thermosphere. The article is well organized and relatively described.
1. The atmosphere-2108575 manuscript addresses the 150-400km atmospheric parameters can be measured using excited gaseous component resonant scatterings. Usually, Rayleigh scattering is used to measure bellow 80km, metal resonant scattering is used to measure 80 to 100km. Because the density is lower and has not enough scattered signals, it is very difficult to measure atmosphere above 100km. The manuscript shows a new resonant scattering can help us to improve the measured altitude. It is exiting news.
2. Do you consider the topic original or relevant in the field? I think it is original in the field.
3. What does it add to the subject area compared with other published material? It has experimentally demonstrated that it can measure 150-400km atmosphere using excited gaseous component resonant scatterings.
4. I suggest that author should research the excited gaseous component resonant scattering more deeply. It will be better if a scattering spectrum provided.
5. The conclusions are consistent with the evidence and arguments.
6. The references are appropriate.
7. The tables and figures are appropriate.
Author Response
Dear Reviewer.
Thanks for your questions
Answers in attached file.

Reviewer 2 Report
The paper presents lidar observation results in 2008–2022. Maybe the work worth publishing, but the paper was not really well organized. It was very difficult to understand. In present form, it is difficult to judge the novelty and significance. So, I suggest the author rewrite the paper and submit it again. I give some examples about the inclarities in the paper, there are much more need to be corrected or explained.
1. In the abstract, the author said: “The appearance of resonant scattering in the atmosphere indicates the presence of ionization sources and does not carry information about the content of atmospheric ions at the ground state.”
The above sentence is not clear.
2. “the working wavelengths of 390.303 nm were proposed for the transmitter and 391.537 nm for the receiver” what was the meaning of working wavelength of the receiver? Because normally, a receiver is composed by a detector and some optics. A more frequency description would be an optical and electronic bandwidth, so the precise wavelength of a receiver need to be explained.
3. “It was found a correlation of the signal with the foF2 critical frequency of the ionosphere [10, 11] at altitudes of 200–300 km.” The sentence was not clear. What is foF2 critical frequency? In Figure 1 b, what is the meaning of the vertical axis? N-Nf, what is the unit of the vertical axis? Is it the strength of lidar returns? The figure caption gave very little information about how the data was collected and presented.
4. Figure 2 need to be explained. The meaning or the definition of the parameters are all missing, it is impossible to understand the meaning of the curves, as long as the authors presented the results to the readers, they need to make effort that the readers can understand what they were talking about.
5. “The state of the ionosphere was controlled by measurements of the “Parus A” 116 ionozonde.”, How can the state of ionosphere be controlled by a measurement?
6. Figure 3, the logic of using double end arrowed lines between PMT2 and PMT1, cnt1 need to be explained.
7. The content in table 2 should be presented with an energy level diagram.
8. The authors should give more explanation about how figure 7 was obtained.
9. the scattering ratio R appeared in Figure 2, but the definition was given only in section 4.3.
Author Response

(The authors gave the same response as above.)
